# Effect of High-Voltage Atmospheric Cold Plasma Treatment on Germination and Heavy Metal Uptake by Soybeans (*Glycine max*)

**DOI:** 10.3390/ijms23031611

**Published:** 2022-01-30

**Authors:** Shikhadri Mahanta, Mohammad Ruzlan Habib, Janie McClurkin Moore

**Affiliations:** Biological and Agricultural Engineering Department, Texas A&M University, College Station, TX 77843, USA; shikhadri@tamu.edu (S.M.); ruzlan277@tamu.edu (M.R.H.)

**Keywords:** soybean, atmospheric cold plasma, germination, plasma activated water, zinc oxide nanoparticles

## Abstract

The need to feed 9.9 billion people by 2050 will require the coordination of farming practices and water utilization by nutrient-dense plants and crops. High levels of lead (Pb), a toxic element that can accumulate in plants, can lead to toxicity in humans. With the development of novel treatment technologies, such as atmospheric cold plasma (ACP) and engineered nanoparticles (NPs), the time to germination and levels of heavy metals in food and feed commodities can be reduced. This study provides insight into the impact of plasma-activated water (PAW) on the germination rates and effects of soybean seeds, and the resultant combination effects of zinc oxide uptake in the presence of lead. Soybean seedlings were watered with PAW (treated for 3, 5, and 7 min at 30, 50, and 70 kV), and the germination and growth rate were monitored for 10 days. The germinated seedlings were then grown hydroponically in a nutrient solution, and the biomass of each example was measured. The PAW treatment that resulted in the best growth of soybean seeds was then exposed to Pb and zinc-oxide nanoparticles (ZnONPs) to investigate heavy metal uptake in the presence of nanoparticles. After acid digestion, the rate of heavy metal uptake by the soybean plants was evaluated using inductively coupled plasma-mass spectrometry. The PAW seeds grew and germinated more quickly, demonstrating that the plasma therapy had an effect. The rate of heavy metal uptake by the plants was also shown to be 5x lower in the presence of ZnONP.

## 1. Introduction

Soybean (*Glycine max*) is the second-largest vegetable oil produced in the world, with the USA being the largest producer [1]. It is among the 16 major crops cultivated worldwide, which makes research on soybean cultivation crucial [2]. Soybeans are consumed worldwide, and various products can be made from the plant, such as soybean oil, soy milk, sprouts, soy nuts, edamame, soy flours, soybean meals, and tofu. It is a good source of protein, fats, and dietary fibers. Soybean has the highest protein content (40–42%) of all food products and the second-highest oil content (18–22%), after groundnuts, among food legumes. Soybean oil is also used as a biofuel and for aquaculture [3]. Soybean, along with maize, rice, and wheat, represents two-thirds of the total agricultural calorie demand [3]. It is projected that soybean production for the year 2050 will increase to 1.3 percent per year, which is not enough to meet the needs of the human population by 2050. Due to the high demand and production of soybean, it is often grown in environments in which the land or water is contaminated with heavy metals. Lead (Pb) is one of the major soil contaminants; it can accumulate in toxic concentrations in seeds when they are grown in a polluted environment [4]. Hence, research was conducted to understand the uptake of Pb by soybean seeds under different treatment conditions.

Atmospheric cold plasma (ACP) is a novel technique that is effective at increasing the shelf life and nutritional value of different biological units. Plasma is created by applying high energy to different gas samples; the gas ionizes and forms smaller species of ions. There are more than 75 ionizing species, neutral species, and UV-visible light, and more than 500 chemical reactions are produced in a plasma system [5]. These ion species react with the different components of the product being treated. The reactive ion species and the ozone produced can generate changes in molecular structures and in their interactions with different biomolecules available in the matrix. ACP also positively impacts the shelf life of different commodities [6]. It reduces the growth of microbes, such as *Escherichia coli*, *Salmonella* sp., and *Listeria monocytogenes* [7]. Due to its low cost and ease of processing, as well as its ability to generate at room temperature and pressure, it can be widely used. The application of ACP in various commodities is being studied; multiple studies have found that ACP significantly increases seed germination [8,9,10]. Ling et. al found that ACP influenced germination when the seeds themselves were treated at 0, 60, 80, 100, and 120 W for 15 s. Fan et. al used plasma at 15, 30, 60, and 90 s and found positive effects on the germination rate, seedling growth, phenolic and flavonoid contents, and antioxidant activity of beans. However, there is still limited research on the impact of ACP-treated water, or plasma-activated water (PAW), on seed germination. 

Lead (Pb) is a toxic heavy metal to which crops may be exposed while growing. Lead contamination in the environment can occur through multiple sources, including natural weather procedures, the disposal of sewage, mining activities, and industry. Lead is also present in products we use daily, such as gasoline, paints, and explosives [11]. When lead is present in the soil and water, it can be easily taken up by plants; high amounts of lead cause toxicity. Lead affects plants in multiple ways, including the inhibition of photosynthesis and mineral uptake, hormonal imbalances, and effects on the cell membrane structure and permeability. Lead can accumulate in the plant and later be passed on to humans as we consume the plant product [11]. With advances in agricultural methods, the large hydroponic production of many leafy vegetables is underway in an attempt to increase the production of consumable plants for the projected population increase [12,13]. The commercial hydroponics market continues to grow and is a viable solution when proper care is taken in preparing seeds and plants to be used in these systems [14]. Plants grown in hydroponic systems have increased bioavailability of organic and inorganic elements [15,16]. Furthermore, plants grown in hydroponic systems have a higher risk of exposure to heavy metals released in the environment [17].

Previous studies have shown that zinc-oxide nanoparticles (ZnONP) have a substantial impact on heavy metal uptake by leafy vegetables and fruiting plants [18]. ZnONPs may promote the growth of plants, increasing the total yield due to the role of Zn in photosynthesis [18]. Furthermore, ZnONPs also have fungicidal properties and may provide up to 80% of the essential nutrients of Zn in plants. Due to the high absorption property of ZnONPs, they may immobilize the free ions of the heavy metals present [19,20]. The higher dissolution rate of ZnONPs in the system may be the reason why heavy metal uptake reduces in the presence of ZnONPs. ZnONPs have been recognized as safe food additives by the US Food and Drug Administration (FDA) [21,22].

The objective of this study was to determine the effects of ACP on soybean seed germination and heavy metal uptake by soybean plants in the presence of ZnONPs. The seeds were initially treated under high-voltage ACP for a short period of time. ACP-treated water was used during the germination and growth of the seeds. The treatment was for a short period of time, making it feasible in an industrial setup. ACP may further increase the solubility of ZnONPs in heavy metal in the plant system. This further reduces the heavy metal uptake by the plants. The study was performed to understand the effects and efficacy of ACP in reducing heavy metal uptake by plants for their sustainable growth. The ACP-treated water used for the germination and growth of the plants can also be effective at reducing any fungal growth that may occur in a hydroponic system.

## 2. Results

### 2.1. Impact of PAW on Seed Growth and Germination

The growth in size of the seeds was monitored to understand the effect of the different treatment conditions in germination of the seeds. The seeds were measured each day for ten days and their growth was compared with that of the controlled seedlings. Most of the seeds germinated in this period and did not grow further in size after the tenth day. The figure below shows the mean size of the seeds during the 10 days. As measured on the tenth day, the PAW with 70 kV treated for 7 min showed the highest mean increase in seedlings, which was an increase of about 26.7% compared with the control group (Figure 1a). A negative value in the growth of the seed size was observed in one of the treatments, as some seeds shrank in size and some died up until the 10th day. The error bars in Figure 1a denote positive standard deviation.

### 2.2. Impact of PAW on Radicle Growth

Effects were observed on the length of the radicles of the seeds with different plasma treatments. About 81.5% of the seeds treated with PAW had longer radicles than the pertinent control groups, with a significance level of *p* < 0.05. The changes in the growth of radicles with respect to the control group (0 mm) were plotted, demonstrating a maximum mean radicle growth of 18.97 mm for the 70 kV, 3 min plasma treatment with PAW compared to the control. A comparison between the lengths of the radicles for different plasma treatments is shown in Figure 2.

The differences in the length of the radicles were visible in the three treatment conditions and they are diagrammatically represented below (Figure 3). In the figure, it can be seen that seeds that were given PAW each day had longer radicle growth.

The changes in the growth rate, germination rate and length of the radicles in the different treatment conditions indicate the effects of PAW on the seeds. The plasma treatment might have generated changes in the affinity of the water molecules to the others. The peroxide levels of the water under different treatment conditions were measured and significant differences were observed. The level of H_2_O_2_ was highest at the maximum voltage and time exposure and the level of H_2_O_2_ was lowest in the water with the minimum voltage and time exposure (Table 1). The changes in the peroxide levels and other chemical changes might have affected the overall growth of the soybean seeds.

### 2.3. Heavy Metal (HM) and Nanoparticle (NP) Accumulation in the Plants

The seeds were grown in Hoagland solution until the first leaves appeared, after which a heavy metal (HM) and nanoparticle (NP) solution was incorporated. After two weeks of growth, the accumulation of the elements differed in the PAW-treated plant samples and the control. The concentrations of heavy metals and nano-particles in the control- and the plasma-treated samples (50 kV, 7 min) were present in variable amounts, which is shown in the table below (Table 2). Inductively coupled plasma-mass spectrometry (ICP-MS) was used to determine the levels of Zn and Pb that accumulated in the plant (Table 2). The soybean seedlings that had been treated with PAW and introduced to heavy metals (Pb) and nano-particles had five times lower levels of Zn (14.229 ppm) and Pb (0.039) accumulated in the plant biomass compared to the control (74.316 ppm and 0.189 ppm, respectively). When grown solely in HM and NP solution, it was observed that the amount of Pb and Zn levels accumulated in the plants were lower than the control samples. The HM uptake of the PAW-treated plants reduced from 3.891 ppm to 0.039 ppm in the presence of NP. A similar trend was observed for the control samples, in which the HM uptake reduced from 4.342 ppm to 0.0189 ppm. A significant reduction of HM uptake by the PAW-treated plants and control plants in the presence of NP was observed.

### 2.4. Biomass of the Plants in Heavy Metal (HM) and Nanoparticle (NP) Solutions

The biomass of the plants differed in both the systems. Furthermore, variation in the biomass was observed under the three different treatment conditions. The weights of the plants were taken initially, and the dry weights were taken after the plants were dried in the oven. Comparisons between the biomasses of the plants in both the systems were performed for the three different treatments, as shown on the graph below. The mean weights of the samples were determined through the one-way ANOVA technique and are shown in the graph below (Figure 4). It was observed that the biomasses of the controlled samples, which were grown solely in the presence of HM, were highest, followed by the samples grown in the solution of HM and NP particles. The biomasses of the controlled samples were lowest for the plants grown in the solution of NP alone. In the PAW-treated samples, it was observed that the plants had the highest biomass when grown solely in HM solution, followed by the plants grown in NP solution. The lowest weights of the PAW-treated samples were observed for the plants grown in HM and NP solution.

## 3. Discussion

### 3.1. Impact of PAW on Seed Growth and Germination

The rate of growth and germination of the seeds varied under different treatment conditions. For the PAW-and-tap-water-treated group, an increase of 23.7% was recorded compared with the control group. However, some shrinkage in the seedlings was observed in the PAW-and-tap-water and the PAW-treated samples. This growth attribute also supports some previous studies conducted on mung bean seed seedling growth with PAW at different plasma exposure times [8]. The growth pattern and shrinkage may have resulted from the enzymatic activity and water absorption in the seeds due to the highly oxidized water chemistry from the activated plasma activated (Figure 5). The acidity and destruction of beneficial microorganisms was also related to this negative growth impact in some studies [23]. The impact of PAW on plant seeds varies depending on the voltage level, the exposure duration, the combination of both voltage and duration, the plant species, the seeds’ original vigor, the feeding gas used, the radicals produced, and other factors. Hence, the inconsistency of the growth output from the PAW treatment is understandable. The growth pattern of the seeds with PAW at 70 kV and 7 min are shown in the figure below.

A more positive effect was reported in a study when plasma-treated seeds of radish (Raphanus sativus), tomato (Solanum Lycopersicon), and sweet pepper (Capsicum annum) plants were used and watered with PAW daily as a combination application [24]. The current study exhibited the positive impact on the seed growth when they were also watered with PAW and tap water. The increase in the germination rate and the germination pattern of the seeds, which showed a positive growth rate, is shown in Figure 6.

Cotyledons were also observed in the seeds.

The seed growth and germination pattern are presented in Figure 6 for the observation days. The increasing pattern of germination in the current study was compared with a study conducted on lettuce germination with PAW [25]. However, the differences between the control group and the PAW-treated group were not significant (*p* > 0.05). The visual cotyledons were recorded at the seventh day of sowing and compared among the treated and control groups (Figure 3). Compared to the present study, a higher germination rate, with a 50% increase, was observed in lentil seeds when the application of He jet produced plasma-treated PAW [10]. The change in the germination also depends on the stimulation of the seed cell membrane and the absence of damage to it by the PAW [26]. In the same study, a low sinusoidal voltage of 13 kV treatment for 13 min direct exposure led to the maximum germination rate of tomato seeds, which contrasted with the current study’s increase in the exposure of plants to PAW treatment. The tendency of the germination rate to decrease in response to the longer application of PAW suggests the complex nature of the time–voltage combination of non-thermal plasma. Optimum conditions for the effective stimulation of the seed germination in the current work were achieved through the 50 kV and 3 min treatment with PAW and tap water.

From the paired sample t-test, for all the voltage groups for different exposure durations, it was determined that the 70 kV voltage was significantly higher than the control group (*p* < 0.05) for seedling growth. However, the 50 kV voltage groups did not differ significantly from the 70 kV (*p* > 0.05) group, while the 30 kV group was significantly different from the control group (*p* < 0.05).

Based on the water application types, the PAW and tap water group had significantly higher seed growth than the PAW-only group (*p* < 0.05), while the results for the seed germination were the opposite. The regular PAW application may have caused oxidative stress on the seeds, resulting in this difference. A well understood treatment level and exposure duration is thus mandated concerning the deep chemistry behind. Nonetheless, a correlation matrix was generated for the impact of PAW on both the seed growth and the germination of the seeds (Table 3). The only positive correlation between both was achieved with the PAW and tap water combination.

Plasma-activated water can interact with the germination simulation process, enhancing the growth and germination rate of seeds [27]. As the discharge voltage, treatment duration and gas feed determine the production of reactive species, the changes in these parameters impacted the seed growth and germination. Since a significant modification from PAW is anticipated but not yet clearly understood, an investigation of the appropriate combination of time and voltage is needed. For instance, longer treatment with PAW may create a highly acidic environment, which may inhibit microorganisms and simultaneously may combine with the oxidation reduction potential of the seeds. A pH of 5.8 to 6.5 did not influence the germination of the *Lactuca sativa* L. [25]. However, a lower pH of 2.7 was also recorded when the seeds were treated with PAW for longer than 5 min [23]. As reactive species are among the major drivers of this signaling of the germination enhancement process, it is imperative to search thoroughly in order to understand the mechanism and impact of PAW on seed growth and germination.

### 3.2. Impact of PAW on Radicle Growth

The current study of soybean seedling radicle growth in response to the application of PAW was compared with the radicle promotion of 51 mm to 55 mm in the *Lactuca sativa* L. seedlings treated with PAW [25]. As a novel application for growth and germination enhancement, some questions need to be answered regarding the biochemical changes to the seedlings in response to PAW. Nonetheless, for 70 kV, a decrease in radicle growth was observed when the treatment exposure was increased to generate PAW. It is also important to understand that radicle growth is stimulated by the water absorption enhancement in response to PAW, as has been evidenced by some studies [8]. Similarly, reactive oxygen and nitrogen species can also regulate seed imbibition by controlling the gibberellic acid hormone and abscisic acid during germination [28]. Changes in the physiological and biochemical components of the seedlings were observed during the experimentation process. The seedlings germinated using PAW each day developed slight pinkish and black pigmentation in the petri dish. This might have been due to the reactive gas species obtained from the PAW.

### 3.3. Heavy Metal (HM) and Nanoparticle (NP) Accumulation in the Plants

It was found that the amount of heavy metals absorbed by the samples was highest when the samples were not treated with the nanoparticles. The amount of heavy metals was the lowest when the samples were treated with only ZnO. Trace amounts of ZnO and Pb were found in all the samples, even when it was not added in the treatment. This trace amount of elements might have been from the Hoagland solution that was initially used as the growth medium for the samples. In the samples that were treated with both heavy metals and ZnO nanoparticles, a higher concentration of ZnO and a lower concentration of Pb were noted. Similar trends were observed in both the treatments. The co-presence of both the elements might have increased their affinity and, hence, their concentration in the samples. However, the amount of ZnO was higher, but the amount of heavy metal absorbed by the plant was significantly lower. This theory is consistent with the aim of the experiment.

All the plasma-treated samples had overall lower concentrations of ZnONPs. The plasma affected the uptake of ZnONPs, which was shown to be due to significantly lower concentrations of ZnO. The plasma treatment might have affected the ZnO affinity of the plants. There was a significant reduction in heavy metal concentrations in the plasma-treated samples. The amount of heavy metals taken up by the plasma-treated samples in the presence of ZnONPs was much lower than the heavy metal uptake by the control samples. The results show the potential for plasma treatment regarding heavy metals and ZnONP uptake by plants. It was seen that the plasma treatment affected the overall growth and element uptake by the samples.

The explicit effects of ZnONPs on heavy metal uptake depend on the specific properties of the nanoparticles and the environmental concentrations of heavy metals. Further investigation is necessary to elucidate whether identical uptake patterns in a soil–plant system can occur. Interestingly, Zn is a key biological inorganic compound in the nutritional properties of soybeans. ZnO nanoparticles may be used in adequate amounts to reduce the heavy metal uptake by plants. Further plasma treatment may be effectively used to understand its effect on the growth of plants, as well as their uptake of ZnO nanoparticles and heavy metals.

Figure 7 shows the changes in seed structure observed from the plants continually watered with PAW (left) and those that were only initially watered with PAW (right). There was major internal damage to the seeds when they were continually watered with PAW, suggesting that some effects or ionization may have occurred. Further biomolecular analysis is suggested to understand the exact effect and mechanisms that occur with this type of treatment. The seeds that were initially treated with PAW and then watered with tap water were used for further treatment and analysis in this work. While some discoloration was noted, the internal structure of the seeds appeared to be the same, as opposed to the results noted on the left of Figure 7.

### 3.4. Biomass of the Plants in Heavy Metal (HM) and Nanoparticle (NP) Solution

In both cases, plants in heavy metal and plants in nanoparticle solution, it was found that the samples containing only heavy metals had a slightly higher weight than the other samples. There were no statistically significant impact of ZnONPs on the mass of the roots. However, the trend was slightly different in the plasma-treated samples. The samples treated with heavy metals also had a slightly higher weight in comparison to the others. ZnONPs had an effect on the dry mass of the plants treated with plasma.

Judging by the biomass of the samples, zinc oxide nanoparticles negatively affected the growth in both the cases. This may be attributed to any toxicity impacts of the ZnONPs or the dissociation of zinc and ions during the growth of the samples. The toxicity of ZnO, which affected the growth of the samples, has been reported in earlier studies, but no direct impact on human health has been found. The extent of the potential adverse effects of ZnONPs on plants directly corresponds to the application of a high level of ZnONPs [29]. The toxic effects of ZnONPs on the growth of soybeans may be a response to inducing the reactive oxidative species (ROS), which leads to genotoxicity and the impairment of the genetic systems of plants [30]. The formation of ROS free radicals can affect chlorophyll synthesis, protein denaturation and genetic expressions [31]. Some studies also show that zinc interferes with the heavy metal uptake of plants. It can compete against the heavy metals as the plants absorb them from the soil/solution.

It was observed that the presence of heavy metals and/or ZnONPs impacted the growth parameters of the plants. The weight of the biomass was higher in the presence of heavy metals, followed by heavy metals and nanoparticles and by nanoparticles in the case of the controlled samples. However, in the plasma-treated samples, the biomass of the samples treated with only nanoparticles was higher than in the samples treated with both nanoparticles and heavy metals. This might have been due to the effect of plasma-treatment, which might have increased the affinity of the plants to the nanoparticles. The change in trend might also have been due to issues associated with the experiment. The overall weight of the controlled samples was greater than that of the plasma-treated samples. The effect of plasma treatment on the biomass of the plants was not recognized.

## 4. Materials and Methods

Plasma-activated water (PAW) was created by placing deionized water in a petri dish inside of a plastic package filled with modified atmospheric air with 5.05% N_2_, 65.03% O_2_ and 29.92% CO_2_. The gaseous combination was used with a higher percentage of O_2_. Due to the plasma treatment, the O_2_ was oxidized to produce O_3_, which helps to generate changes in the treated products, along with the other reactive gas species produced in the system [6]. The package was then placed in the ACP system (Figure 8), which creates a non-equilibrium plasma between two electrodes, inside of a package. A voltage was applied from the outside of the sealed package using a high voltage supply and variable controller. Initially, we treated water samples of 100 mL at 30, 50, and 70 kV for 0, 3, 5, and 7 min [32]. Once the treatment was complete, soybean seeds were submerged in the water for 25–30 min inside the package. *Glycine max* (L.) (soybean) seeds were purchased from Johnny’s Selected Seeds (Winslow, ME, USA). The moist seeds were then carefully placed on paper placed over a petri dish. The seeds were arranged as 5–6 seeds in each dish to avoid overcrowding and ensure uniform growth. The H_2_O_2_ levels of the PAW sample were checked and the levels of H_2_O_2_ in different treatments are shown in the table below.

### 4.1. Process Optimization

The effectiveness of the atmospheric cold plasma (ACP) method was confirmed by analyzing methylene blue (MB) discoloration. Methylene blue is highly oxidative and reduces when ionized. This leads to a change in the dye color. Analysis was performed according to the processes described in McClurkin et al., 2017 [6]. The color of methylene blue is dependent on the voltage treatment and time exposure; the rate of color changes is higher with more oxidation.

### 4.2. Plant Culture and Exposure Experiment

#### Germination Process

Germination of the soybean seeds were performed under controlled conditions at 75% RH, 27 °C, and 12 h light. The seeds were initially immersed in PAW and deionized water for 40–45 min. The wetted seeds were then transferred to a petri dish and covered with a filter paper. Next, 5–6 seeds were placed separately in each petri dish to avoid overcrowding during their growth. The plates were placed inside an incubator under the above-mentioned conditions. Approximately 2–3 mL of water was supplied to the seeds each day during the germination process. The plates were rotated within the incubator to ensure the uniformity of the samples. After 4–6 days, the seeds germinated, and the lengths of the radicles were measured. When the length of the radicles grew to >2 cm, they were properly washed with distilled water and transferred to the next stage.

### 4.3. Growth Process of Soybeans with Nanoparticles and Heavy Metals

The germination stage was followed by the growthof the seeds in the Hoagland solution. Hoagland solution of 50% strength was prepared, sterilized, and cooled down. The pH of the solution was measured. Sterile 50 mL polypropylene tubes were taken and filled with Hoagland solution. The tube was then covered by aluminum foil and a small hole was formed on the top. The seed was placed on the top of the tube and the radicle grew inside the tube, within the Hoagland solution. The seeds were kept under the same growth conditions as mentioned above. The amount of Hoagland solution uptake by the seeds was monitored each day. Initially, the amount of water uptake was low, but it increased as the seeds grew, as expected. The amount of water increased to ≥10 mL when the seeds were fully grown, after 2 weeks. The seeds were allowed to grow for 14–16 days in the Hoagland solution.

The seedlings were ready to be transferred to the next stage. In this stage, the Hoagland solution was replaced by heavy metal, nanoparticles, or a combined heavy metal and nanoparticle solution. Negatively surface-charged ZnO NPs (20% by weight, 10–30 nm) were purchased from US Research Nanomaterials, Inc. (Houston, TX, USA). High-purity Pb (Pb(NO_3_)_2_ > 99%) was obtained from Alfa Aesar (Ward Hill, MA, USA). The plants were properly washed and transferred into new 50 mL polypropylene tubes. Four different sets of solutions were prepared. The tubes were filled with solutions containing 100 mg L^−1^ Pb^2+^ and/or 100 mg L^−1^ ZnO-ENPs in DI water. Replicates of three were performed for each sample, and blank controls were grown. During the exposure period, the Hoagland solution was avoided, since the high ionic strength of Hoagland solution that induces the aggregation of ZnO-ENPs. The growth condition was maintained as before.

### 4.4. Drying and Acid Digestion

The plants were ready for drying after growing in the solutions for five days. The plants were then removed from the hydroponic system. The plants were first washed with 50 mL CaCl_2_ solution (5.0 mM) five times to remove any Zn or Pb elements that might have deposited on their surfaces. The samples were then rinsed with distilled water three times. The fresh weight of each plant was obtained, and the plants were kept in the oven for drying. The plants were dried at 75 °C for 48 h. After complete drying, the dry weights of the plants were determined. The process of acid digestion was conducted in accordance with Sharifan et al., 2020 [18].

### 4.5. Heavy Metals and Trace Elements of Content Analyses

For the digestion procedure, 3 mL of 70% (*w*/*w*) nitric acid (certified ACS plus) was added to 0.08 g of biomass in a 15 mL digestion tube. The dried plants were then mashed using a mortar and pestle. Acid digestion was carried out for 120 min in a hot block digester. After cooling down, 2 mL of 30% (*w*/*v*) H_2_O_2_ was added for a clear solution. To determine the quantity of nanoparticles and heavy metals in the plants, the acid-digested samples were further diluted to five times using DI water. The element concentrations were quantified by an inductively coupled plasma-mass spectrometry ICP-MS (Agilent 7500i, Agilent Technologies Co., Ltd., Santa Clara, CA, USA). The ICP-MS method of heavy metal detection was conducted in accordance with previous experiments by Islam et al. [33].

## 5. Conclusions

The seedlings and germination of soybean seeds can be affected by PAW treatment, although the combination of voltage and treatment duration requires further studies to determine its optimum output. The growth of the radicles was greater in the seedlings treated with PAW each day; this could potentially reduce the germination time. However, some changes in the seed structure were observed in the PAW seedlings.

Significant differences in the amount of heavy metals and ZnONPs were observed in the samples. There were also variations in the amount of element uptake in both cases. The mean dry weight of the biomass was higher for the controlled samples compared to the plasma-treated samples. The total heavy metal uptake and nanoparticle uptake were also higher in the controlled samples.

## Figures and Tables

**Figure 1 ijms-23-01611-f001:**
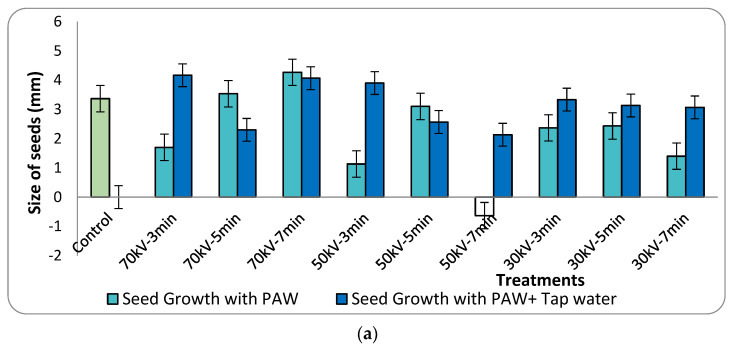
(**a**) Growth in the size of the seedlings on the 10th day at different treatment conditions. (**b**) Number of seeds germinated on different selected days under different treatment conditions. After 10 days of sowing, about 60% to 93.3% of seeds were germinated. The maximum germination of seeds was 93.3%, observed in the 50 kV 3 min- and 50 kV 5 min-treated PAW and tap-water-applied group (**b**) which was 7% higher than the control group.

**Figure 2 ijms-23-01611-f002:**
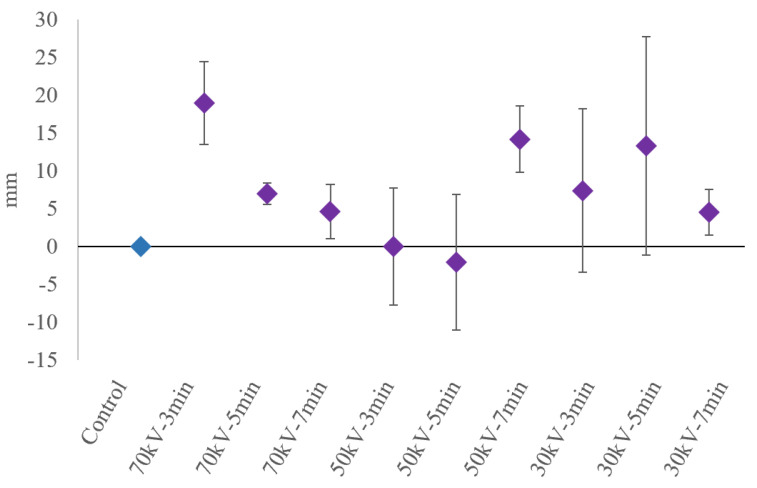
Mean radicle growth changes in the seeds with PAW from the control group (error bars represent standard deviation from the mean).

**Figure 3 ijms-23-01611-f003:**
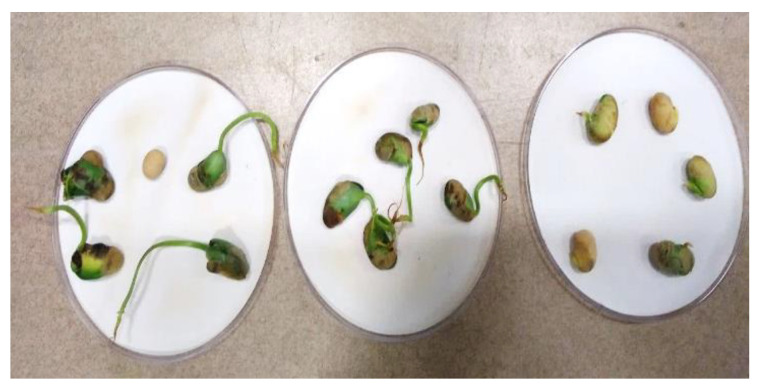
Day 7, when radicle growth was >2 cm. Left to right: PAW daily treatment, initial PAW treatment and control.

**Figure 4 ijms-23-01611-f004:**
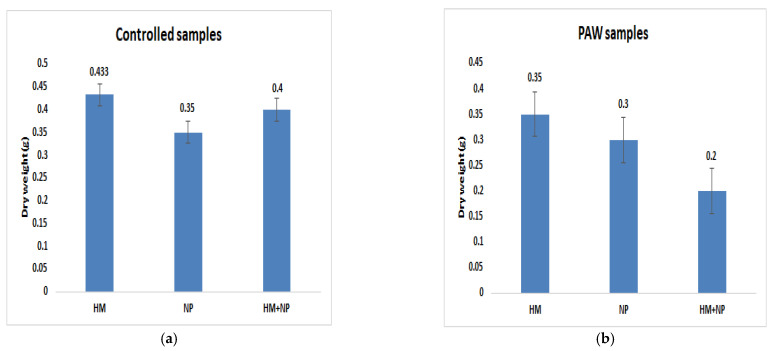
Dry biomass of *Glycine max* (L.); combinations of 100 mg/L Pb and/or 100 mg/L ZnO-NPs. (**a**) Control samples with different treatment. (**b**) Plasma-treated samples with different treatments. The values in the columns represent mean values of the samples and error bars.

**Figure 5 ijms-23-01611-f005:**
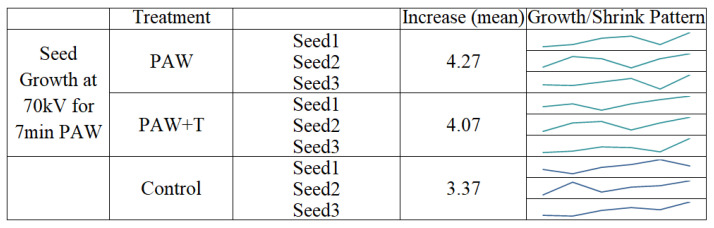
Daily mean growth of seeds after 10 days with 70 kV, 7 min PAW. The growth/shrinkage pattern of the seeds is also shown.

**Figure 6 ijms-23-01611-f006:**
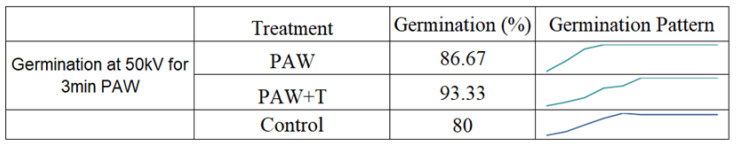
Germination percentage and the growth pattern of seeds treated with PAW at 50 KV for 3 min and PAW + TW at 50 KV for 3 min.

**Figure 7 ijms-23-01611-f007:**
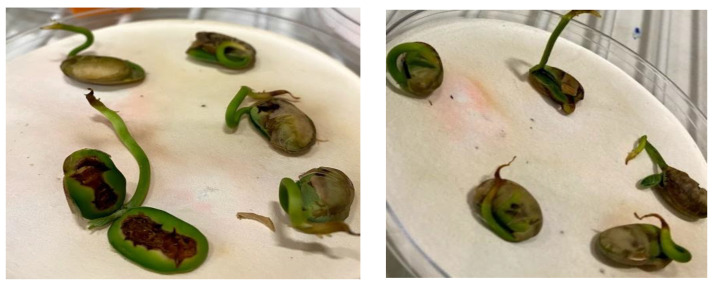
Changes in the seeds’ structure after treatment with PAW (**left**) compared to the PAW + TW (**right**)-treated seeds on 6th day of germination.

**Figure 8 ijms-23-01611-f008:**
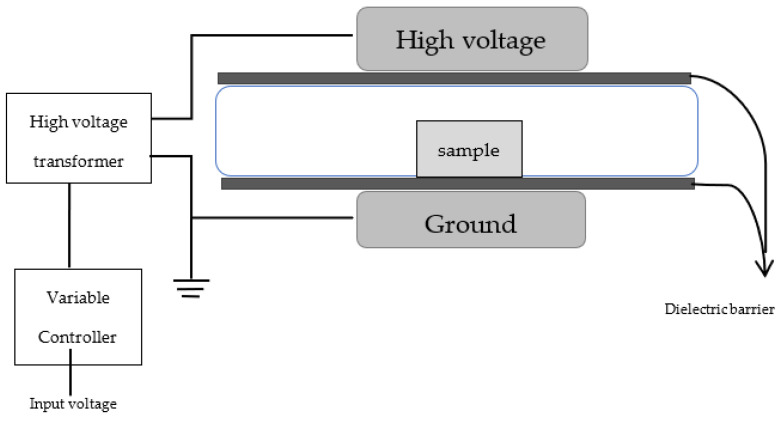
Schematic diagram of a high-voltage atmospheric cold plasma system. Note that image is not to scale.

**Table 1 ijms-23-01611-t001:** H_2_O_2_ level of water at different rates of plasma treatment.

Treatment	Level of H_2_O_2_ (ppm)
70 kV, 7 min	>100
70 kV, 5 min	>100
70 kV, 3 min	100
50 kV, 7 min	100
50 kV, 5 min	10
50 kV, 3 min	2
30 kV, 7 min	2
30 kV, 5 min	1
30 kV, 3 min	1

**Table 2 ijms-23-01611-t002:** Heavy metal (HM) and nanoparticle (NP) accumulation in control (C) and plasma-activated water (PAW) samples.

Samples	HM		NP		HM + NP	
Treatment	C	PAW	C	PAW	C	PAW
Zn (ppm)	16.905	2.672	40.061	16.848	74.316	14.229
Pb (ppm)	4.342	3.891	0.017	0.041	0.189	0.039

**Table 3 ijms-23-01611-t003:** Correlation matrix of the seed growth and germination increase in response to treatment with PAW and tap Water (TW).

		Growth Increase		Germination Increase	
		PAW	PAW + TW	PAW	PAW + TW
Growth increase	PAW	1			
	PAW + TW	0.218	1		
Germination increase	PAW	−0.368	0.182	1	
	PAW + TW	−0.384	−0.341	0.063	1

## Data Availability

Data supporting the reported results can be found in the Texas A&M University OakTrust Data Repository (Mahanta et al., 2022).

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
