# Peer review of "Effect of High-Voltage Atmospheric Cold Plasma Treatment on Germination and Heavy Metal Uptake by Soybeans (*Glycine max*)"

_ijms, 2022, doi:10.3390/ijms23031611_

Round 1
Reviewer 1 Report
The manuscript entitled “Effect of High Voltage Atmospheric Cold Plasma Treatment on Germination and Heavy Metal Uptake by Soybeans (Glycine max)” deals with the impacts of plasma activated water on the germination rates of soybean seeds. Moreover, it analyses the effects of zinc oxide nanoparticles and lead in combination with plasma activated water on soybean seedlings. As nature pollution by heavy metals and the potential of plasma activated water use in agriculture are very actual, I consider the aim of the manuscript very topical.
However, the potential of this study has not been fully realized. I think authors should have investigated problematics in more detail to allow proper conclusions to be drawn. Many publications have shown effect of plasma or plasma activated water on the seed germination including soybean. This paper absolutely needs to be reworked and analysed in depth before being resubmitted.
Abstract
- Authors mentioned that they used exposure times 1, 3 and 5 minutes for PAW production in their study in abstract, however they really used exposure times 3, 5 and 7 minutes.
Results
- Controls in Figure 1a are expressed by blue/green bars, which according to the figure caption means that they are treated with PAW/PAW + tap water. How were controls for this experiment provided?
- Which parts of seedlings were measured in the Figure 1a? How is it possible that value for sample “50 kV – 7 min” (green bar) is negative (in millimetres) in Figure 1a?
- How is it possible that length of some parts of analysed samples “7 min – 70 kV PAW” and “3 min – 50 kV PAW” in Figure 1b are shorter in days 8 and 10 compared to day 6?
- Figures 1a and 1b should be properly marked inside the figures.
- There is missing statistical analysis (such as ANOVA or t-test) in the Figure 1a, 1b, 3, 4 and 5 and Tables 1 and 2 (Tables 1 and 2 have no standard deviations).
- How many times were experiments for Figure 1b repeated? Why are there no standard deviations in the Figure 1b?
- Figure 1b has not the right caption and has not marked vertical axis.
- I do not understand Figure 2. What authors want to show by implementation of this figure in the manuscript?
- Figures 2, 5 and 6 and Table 2 are not commented in the Results of manuscript.
- I do not understand the Figure 5, figure caption is insufficient.
Discussion
- Results are not sufficiently explained and confronted with existing literature.
- Changes in the germination of samples are not explained. Why there is no observable trend in germination dependent on exposure time or kV? Analyses of the activity of germination enzymes will be improvement for the study.
- Authors do not provide clear explanation for pigmentation changes of seedlings after PAW treatment. These changes should be investigated more in depth by various biochemical methods.
- Authors do not sufficiently explain why there is a decrease in zinc content samples treated with PAW compared to control that results from Table 1.
Methods
- Authors do not cite literature in the most of methods. Do they develop these methods by themselves?
- It is not clear what is the difference between PAW treated and PAW + Tap water treated samples.
Formal errors
- Title Dr. should not be included in the name of author (Dr. Janie McClurkin Moore).
- Citations in the text are not from the lowest to the highest number.
- Authors incorrectly use “seeds” in text about “seedlings” (for example, authors measured radicles of seeds in Figure 3).
- Authors do not use italics for Latin names of plants in the almost entire manuscript.

Author Response
Thank you

Reviewer 2 Report
In the manuscript, the authors report an effect of plasma-activated water (PAW) on germination and heavy metal uptake of Soybeans. Consequently, the authors shows a higher seed growth and germination in PAW treated seeds than those of control. In addition, they present that PAW reduces the lead (Pb) uptake in soybean plants. This work represents interesting opportunity for the PAW’ effect to the agriculture field. However, some issues need to be addressed before processing further. My other comments are as follows:
- In the abstract, the explanation on experimental results seem to be insufficient. It would be more helpful to understand if final experimental value (point) and its meaning are shortly mentioned.
- It is hard to understand the result of Figure 1. What is the control? Dose the control group indicate Tap water only?
- In the Figure 1, I am wondering why you compared PAW group and PAW+ Tap water, not Tap water group (plasma-untreated water) and PAW group.
- In the Table 1, it was found that PAW-treated sample (2.672 ppm) remarkably reduced the level of Zn when compared to control (16.905 ppm), whereas it is likely there is no big difference in the Pb level between both groups (3.891 and 4.342, respectively). The authors should address it.
- The authors mentioned the levels of pH and H2O2 of the PAW sample in the Materials and Methods. Because their levels would be important to address the experimental results, it should be shown as a figure in the manuscript.
- In the Materials and Methods, PAW was created inside of a plastic package filled with modified atmospheric air with 5.05% N2, 65.03% O2 and 29.92% CO2. I am wondering why the authors used the modified air instead of normal air.
- The color change of methylene blue was analyzed for the effectiveness of the ACP. Is the color change of the methylene blue dependent on the time and voltage of plasma discharge?

Author Response
Thank you

Round 2
Reviewer 1 Report
Dear authors,
Thank you for the answers and for implementation of my comments to your improved manuscript.
I think that it will be better to move Figures and Tables from the Discussion chapter to the Results. However, it is maybe just my opinion.
Best regards
Reviewer
